# Creating Family-Centred Support for Preschoolers with Developmental Disabilities in Low-Income Countries: A Rapid Review to Guide Practitioners

**DOI:** 10.3390/ijerph21060651

**Published:** 2024-05-21

**Authors:** Roy McConkey

**Affiliations:** Institute of Nursing and Health Research, Ulster University, Belfast BT15 1ED, UK; r.mcconkey@ulster.ac.uk

**Keywords:** disability, preschool, children, family-centred, community-based, low income, inclusion, healthcare, education

## Abstract

Preschoolers with disabilities and their caregivers have been neglected in health and social service provision in most low-income countries and arguably also in low-resourced areas of more affluent nations. Yet as this rapid review of the published literature identifies, there are low-cost, evidence-based strategies to address their needs that can be implemented in communities by local people. Five key features of the necessary supports are examined. First, the leadership functions required to create and implement the support services. Second, the family-centred, home-based support provided to caregivers and the personnel undertaking this form of support. Third, providing opportunities for peer support to flourish and encouraging the formation of advocacy groups across families. Fourth, mobilizing the support of significant groups within the community: notably, traditional healers and leaders, health services and poverty alleviation initiatives. Fifth, devising ways in which preschool educational opportunities can be offered to children as a prelude to their inclusion in primary schools. The review serves a further purpose. It provides an example of how public health researchers and academics could achieve more rapid implementation of evidence-based knowledge into existing and new support services through dissemination to community practitioners.

## 1. Introduction

Preschoolers with developmental disabilities face a bleak future around the world. Those living in low-income nations are particularly disadvantaged. But those born in low-resourced communities in richer countries also face a precarious future. Yet, it need not be so. In this review of empirical research, my aim is to capture the practical steps that can be taken by practitioners in any or every country to enhance the life chances of youngsters with developmental disabilities, primarily through family-centred, family-driven supports.

Nor is this intended to be an academic endeavour. Rather my message is for practitioners—be they family members, community workers, health professionals—who are often eager to assist but feel they lack the knowledge and skills to do so. I acknowledge that few such practitioners may read this article. Rather, my hope is that researchers and educators in our universities, higher education colleges, and health and social services will use the content in their training courses, thereby setting off a dissemination pyramid that will reach families and communities in some, maybe many, nations.

This review of reviews may also serve as a modest example which public health researchers and academics could follow to provide more rapid implementation of evidence-based knowledge into existing and new support services. Admittedly, this is a complex enterprise of which documentary reviews are but a start. Nonetheless, they can be valuable, not just for the guidance they offer for practitioners, but also by pointing out directions of travel for future research.

Appendix A gives further details of the process followed in creating this review of reviews as well as listing the freely available resources and articles cited in it. This is the first of two articles; a second one describes how family-centred support services have been developed in low-resourced communities in Zimbabwe [1].

## 2. The Art of the Possible

When faced with complex issues, the temptation is to start a review by describing the many difficulties that need to be overcome. At best, such an analysis may provide a road map for tackling the challenges but at worst, listing the difficulties may discourage any attempt to overcome them. Meeting the needs of preschoolers with developmental disabilities in low-income countries and low-resourced communities easily qualifies as a complex and challenging endeavour. Within global health, a favoured strategy for dealing with such issues has depended on high-level policy formation and its implementation, but the approach has had mixed results for disabled persons [2]. Instead, what if our approach was centred around ‘grass-root’ responses to the needs of persons with disabilities? That is the core premise of this review, based on the presumption that local people—even in disadvantaged communities—have assets available to them so that they can start to tackle the difficulties they face [3]. This approach not only offers hope that solutions can be found but the identified assets provide the building blocks around which implementation actions can be created. This might be called: the art of the possible.

Five main assets are available to support families who have preschoolers with disabilities.
There has been an accumulation of knowledge and skills over the past 20 years as to how preschoolers with disabilities—those aged 0 to 6 years—can be helped [4,5]. These have come from within and across nations, especially in richer countries but also countries with fewer resources. We can confidently claim that the development of all preschoolers can be enhanced, and we can specify practical ways of doing this.Families are the main caregivers and educators of preschoolers all over the world. In low-income countries, often they are the only ones available to assist boys and girls with disabilities. Hence, parents are the bedrock for creating better lives for disabled preschoolers. Most mothers and grandmothers rise willingly to the challenge, as will many but not all fathers. Hence, support for the child needs to be delivered through support for their family carers and it has been shown to be effective [6].Even within the most impoverished communities, families have found ways of coping with adversity while demonstrating remarkable resilience. Moreover, they can draw upon their indigenous knowledge and cultural heritage in rearing their children, allied with the support of their relatives and local community [7]. Discovering ways of mobilising family and community supports creates an essential asset.People of good intent are present in every community. Their motivation to assist others in need, allied with knowledge of what practical steps they can take, is a powerful contributor to success. Among those who might instigate support for families who have a preschooler with disabilities within local communities could be parents of older children with disabilities, unpaid and paid community workers, community and religious leaders, professionals such as therapists, teachers, social workers, and local politicians and government officials. In truth though, it could be anyone with a heart to help their fellow citizens and the confidence to approach others for the cause. Identifying likely and available leaders of new support services is a key ingredient of creating change [8].Admittedly, the impetus for creating community-based services and supports for persons with disabilities has mostly come from outside the local community, albeit with the intention of sharing ownership of the project with leaders from that community. This external form of leadership has tended to come from national and international Non-Governmental Organisations (NGOs), Disabled Persons Organisations, or Parent-led Associations, but also from front-line health and social services and educational professionals [9]. Arguably, some of these external supports are or could be present in nearly all communities in all countries. Supports for families of preschoolers with developmental disabilities may not feature at all in their current workload and few may be currently motivated to include them, but some will be prepared to do so. This article may also serve to challenge and inform practitioners in these external organisations as to how they might instigate and support local community responses for this marginalised group of families.

## 3. An International Imperative

But why would any of the organisations noted above as well as local communities be bothered to help preschoolers with developmental disabilities? Four core reasons can be marshalled.
International statements on rights of children and of persons with disabilities have been mostly incorporated into national laws and policies. They emphasise that discrimination based on disability is unacceptable if not unlawful. For example, Article 23.3 of the Convention on the Rights of the Child states: “*Recognizing the special needs of a disabled child, assistance … shall be designed to ensure that the disabled child has effective access to and receives education, training, health care services, rehabilitation services, preparation for employment and recreation opportunities in a manner conducive to the child’s achieving the fullest possible social integration and individual development, including his or her cultural and spiritual development*”. (last accessed on February 2024).There has been a shameful lack of progress in responding to the needs of preschoolers with disabilities. A recent review undertaken by the Collaboration for Global Research on Developmental Disabilities [10] noted that over 53 million preschoolers globally have a developmental disability with an overall prevalence rate of 7.5% among under fives. Yet, an analysis of the funding provided by Development Assistance for Health to low- and middle-income countries [11] which totalled USD 76.3 billion during 2009 to 2016 found only USD 0.7 billion (2% of the total) was for disability. Moreover, this funding declined over that period despite an increasing prevalence rate [12]. Consequently, governments seemingly have neither requested nor been given or found the financial resources to address the needs of these preschoolers. This neglect is unjust.These preschoolers and their families experience multiple disadvantages, chief of which is poverty allied with poor physical infrastructures, such as housing, sanitation, and transport [13,14]. Responding to these needs arguably could bring benefits to other households experiencing similar difficulties in the community, especially as the empowerment of local people is a central response in meeting these needs, as will become apparent in later sections. Strategies to help a few can result in many benefitting.Creating educational systems that are inclusive of pupils with disabilities is an international imperative to achieve the sustainable development goals agreed by the world’s nations. For example, Sustainable Development Target 4.2 requires that by 2030, “*all girls and boys have access to quality early childhood development, care, and pre-primary education so that they are ready for primary education*”. (http://www.un.org/sustainabledevelopment/sustainable-development-goals: accessed on 1 February 2024). The foundations need to be laid in the preschool years both in terms of boosting parental engagement in the child’s education and through the provision of early childhood education centres. Yet, fewer than 30% of preschoolers in many African countries, for example, attend such centres [15].

### Common Causes of Inaction 

It would be foolhardy to ignore the reasons for why preschoolers with disabilities have been neglected in the delivery of early childhood services, especially in low-income countries. Three inter-related causes have been identified in past research.
The stigma around disability persists in many communities which has created shame in families and an unwillingness to request assistance. Likewise, parents have been shunned and faced ridicule even from their own relatives. Well-tested strategies have been identified for reducing stigma [16], most notably through the building of personal relationships among the affected families and in turn with their community neighbours. More positive attitudes result in increases in the child’s social inclusion [17].The lack of knowledge within families and community services on how best to help the child manifests in feelings of helplessness and hopelessness, and ultimately in the exclusion of the child from health services and preschool education. A necessary response is the provision of learning opportunities to families and to community personnel, allied with their personal experience of seeing the children progress [11].Preschoolers with disabilities are not a priority in low-income countries when so many other groups also have unmet needs [18]. Nevertheless, new or extra resources are not necessarily required by these families or the children. Rather, supports provided to other groups should be made inclusive of their needs, be it for income generation, improved housing, and transport [19]. But for this to happen, the above two points need to be addressed.

None of these issues will be speedily addressed and indeed they may never be fully realised in some communities despite the dedicated attempts to do so. However, as you will see, these three aspects must be incorporated into the strategies used to create family-centred supports for preschoolers with disabilities, which is the focus in the following sections.

## 4. Making a Start—The Role of Service Leaders

There is no one blueprint for creating supports for families. The approaches used can vary according to the agency and people involved in planning and delivering the services, be it community health workers, preschool educators, or parent associations. Also, the existing resources within a community will have an influence; for example, a primary school that has started to include pupils with disabilities may use different strategies to a setting where community facilities are sparse. That said, in this and subsequent sections, details are given on common strategies that leaders of service developments may use to a greater or lesser extent and for which evidence exists of their effectiveness. Figure 1 provides an overview of them.

### 4.1. Leadership

The term ‘leader’ is used for an individual or small group of persons who, in this instance, are motivated to develop supports for families of preschoolers with disabilities and who take responsibility for designing and implementing the actions required to instigate and sustain the supports. As will become apparent, the hallmarks of their leadership style are the building of partnerships and shared decision-making.

At the outset, four strategies can be used to demonstrate this style of leadership as well as laying the foundations on which the operations of the support services can be built. These will take some time and effort, but they are crucial to the sustainability of the endeavour.

#### 4.1.1. Identify Allies from Within the Community 

A valuable asset is the recruitment of people from the community and from existing service providers to join a steering group for the support service. They will provide insights about the community, its structures and influencers, and make introductions for you. They can also help to share the workload and build a sense of community ownership of the proposed service. As the service evolves, other members can join the group, notably some parents of preschoolers.

#### 4.1.2. Determine the Resources Available in the Community

What other services and organisations are active in the community and how are they funded? What facilities are available, such as community halls, transport, and faith communities? Who are the leaders within the community—local chiefs, traditional healers, local government politicians and officials? What is working well in the community and what community development projects have been tried in the past but failed? Community conversations offer a useful approach to answering such questions [20].

#### 4.1.3. Assess the Needs of the Intended Beneficiaries

This can be achieved by speaking with a small number of parents, perhaps contacted through word of mouth. More ambitious would be to undertake a series of meetings with groups of parents and involve them in planning the supports that would assist their child and well as their personal needs and those of the family as a whole. Participatory Action Research has shown how this can be done [21]. Such planning groups could be the basis on which peer support networks can be built. Ideas are given later for achieving this.

#### 4.1.4. Evolve an Action Plan for the Support Service

The foregoing information can be summarised in an evolving plan that outlines the main aim and intended goals of the support service, the main activities that will be undertaken and who will deliver them, the other resources available to the service and expected outcomes for the child, and the family and community [1]. Variations of this plan can be used to publicise the service to prospective users, to inform the wider community along with other service providers, and to seek funding from donors and governments.

## 5. What Is Meant by Family-Centred Supports?

The family, and especially mothers, are critical to the development of preschoolers with developmental disabilities. Hence, the recent literature emphases the need to create family-centred and community-based support services rather than focusing only the child’s disabilities, which has been the dominant approach across the globe in past decades, undertaken by clinical and hospital-based medical or rehabilitation services [3].

Moreover, the attributes of family-centred/family-driven supports have been identified. These are based on key principles and values that are expressed in building respectful and trusted relationships between support workers and parents [22,23,24] to which can be added respect for family cultures, values, and beliefs, which is especially pertinent in non-Western countries and for immigrant families from these cultures [25]. These include: Sharing knowledge and information with parents so that families can make informed decisions.Respect for families’ culture, beliefs, and values.Recognising and building on family members’ strengths and resources, and active participation by family members in assessments of the child’s needs and developmental interventions.Psycho-social supports to enhance mothers’ emotional wellbeing and increase their feelings of self-advocacy.Encouraging advocacy by families in support of their rights.Providing or mobilising supports and resources in response to family concerns and priorities.

All these ambitions can be hard to realise from the outset. Rather, they are likely to evolve as the service develops and families grow in confidence, realising that they can make change happen. This has been termed ‘self-efficacy’ and has been increased with parents involved in intervention programmes, especially those with children under five years of age [26]. A crucial element in nurturing self-efficacy is the nature of the relationships forged between the family and their support workers. The words used by parents to describe the qualities of supporters they found helpful were ‘trustworthy’, ‘reliable’, ‘good listeners’, and ‘treated parents as equals’ [27]. Recruiting family supporters with these personality traits is arguably more important than their past training and qualifications.

Often, claims are made that support services are family-centred when this may not be so, such as when therapists prescribe programs for parents to carry out at home or preschool staff organise training courses for mothers [28]. Self-assessment scales are available that services can use to design and review the extent to which they practice the values and core activities of family-centredness described above [29].

Reviews of evaluations made of family-centred supports have confirmed the benefits they have achieved for children and their caregivers [30]. These include: children acquiring new skills, family needs being met, better quality of life, increased parental satisfaction, and greater community involvement.

## 6. Creating Family-Centred Support Services

Family-centred support is best delivered in the family home, at least at the outset, but it should continue even when other services become available, such as the child attending a creche, nursery, or school, albeit by personnel from the centres visiting the family home. Research has shown better outcomes are achieved for the child and families when this happens [19].

The benefits of personnel visiting the home are that they can meet all the family members, and they can observe their living arrangements and their interactions with the child as well as familiarising themselves with the immediate community in which the family lives. The visits remove any transport difficulties for the parents. They provide a more relaxed meeting that gives time for relationships to develop. Moreover, the advice and guidance provided to parents can be adjusted to the family situation, using whatever space and materials that are available [31].

However, professional staff such as therapists rarely have the time and opportunity to undertake home visits, desirable as that would be. Rather, the visits are mainly undertaken by ‘first level’ community workers [32]. Among those who have taken on home visiting roles in low-income countries are community health workers (paid and unpaid), community nurses, community-based rehabilitation workers, preschool personnel, and teachers and volunteers from the local community, including parents of children with disabilities. Most are likely to require basic training in how to assist preschoolers with a disability alongside promoting a family-centred ethos in their visits. Thus, leaders play a vital role in both the recruitment and training of family support workers.

### 6.1. Recruiting Family Support Workers

Various strategies have been employed to recruit workers from the community [33]. The leaders may nominate persons known to them who they feel have the personal qualities and motivation to take on the role. If there is a Disabled Persons Organisation or a Parent and Friends Association, they can be asked to identify persons. Advertising posters can be placed in clinics and community facilities inviting people to apply and alerting them to the training which will be available. A public meeting can be held with an open invitation to all interested persons or visits made to community groups such as faith communities to inform them about the proposed service and how they might support it. Hopefully, more people will express an interest than are required.

### 6.2. Basic Training and Selection

Leaders have invited interested persons to come along to a meeting when more details are given about the aims of the project and why it is starting in their community. The role of the family support worker is explained, and attendees have the chance to meet some parents and hear more about their children with disabilities. Their questions about the project can also be answered. They are informed that training will be provided and given a date when this will commence. Those who decide not to become involved can withdraw gracefully by not coming to the next meeting.

The training might take place over several sessions, depending on the amount of information and skills training that the leaders judge is required to launch the home visits while recognising that there will be an ongoing need for continual, ‘on-the-job’ training and supervision. Fortunately, there are various training manuals and packages available that focus on early child development with children who have disabilities, although they may need to be translated into local languages. Most have been designed for use with parents but equally they can be adapted to prepare home support personnel [1,34,35]. They describe the nature and causes of disabilities, the concept of development, and the play-based activities that will enable children to learn new skills (further details are given in subsequent sections). One important additional topic not always covered is the nature of the interactions that support workers have with family caregivers as described above.

Throughout the training, the use of practical activities—rather than giving talks—is recommended so that leaders can assess which participants demonstrate the qualities they are looking for in support workers and their motivation for the role. Examples are given below.

The basic content has been delivered in various formats, such as two one-day sessions or in six to eight two-hour sessions over the course of two months. Again, some people may drop out over the sessions, but they leave more aware of children with disabilities and how they can be helped.

Those who complete the course can then be invited to apply for whatever number of posts that are available depending on the funding available for their salary. Unsuccessful but suitable persons can be encouraged to maintain contact with the project through undertaking other voluntary roles and the promise of future work if the project expands.

### 6.3. Ongoing Training and Support

The family support workers will need ongoing support to further develop their skills and responses to the family needs. This is usually provided by the project leaders who may meet regularly with them to review their work with families, often best conducted by accompanying the family support worker on occasional home visits. Regular review meetings or telephone contacts have also been used.

Support workers might also accompany the family to therapy sessions in clinics or hospital appointments. This too can provide them, as well as the caregiver, with further advice and guidance on helping the child acquire new skills which can be used by caregivers at home.

Further training workshops, lasting up to three hours, can be organised for home visitors and family members on specific topics. They may also be sponsored to attend courses organised by other services locally or in neighbouring towns and cities.

## 7. Home Visits

Visits to the family home tend to be more frequent at first but taper down over a period, starting every two weeks, then monthly and bi-monthly. They tend to be for 60 to 90 min. Telephone contact has also been used between visits. However, there is a trade-off to be gauged between the number of families to be visited, the number of home visitors available and the number of days they work, the number of visits per day (taking into account travel time), and the supervision that leaders can provide. Common sense suggests it is better to start off with a small number of families and increase gradually in light of local experience and available resources. Limits may also have to be set on the length of time families remain with the project (for example, to when the child is six years old) so that ‘new’ families can join.

How the visitors travel to the family homes will need attention: walking, use of local minibuses, and the provision of bicycles or motorcycles have all been used, but cars and four-wheel drives are only available in the best-resourced projects and their high running costs are often unsustainable once international aid finishes.

The content of each home visit will be dictated by the needs of the child and caregiver but in broad terms, three elements will likely feature.

### 7.1. Identifying the Child’s Strengths and Needs

Early visits will be used to assess the children’s development—what they can do and what they cannot do. Different tools are available that family support workers can use with caregivers. For example, UNICEF have developed a series of questions for obtaining a quick assessment of children aged 2 to 4 years of age [36]. This was developed to screen children for disabilities in household surveys and is available in many different languages.

More detailed developmental assessments can be made using other tools designed for children in LMICs [37,38]. These examine the child’s development in different domains: notably, motor, communication, social, personal care, and cognitive skills. The advantage of these scales is that they identify the steps that come before a desired developmental goal will be attained, such as the child walking and talking.

During a home visit, the support worker will assist the caregiver in completing the scales both through conversation and also by observing and ‘testing’ the child at home. To assist families, simplified versions of the scales have been developed in local languages, using fewer items and mainly pictorial symbols to overcome literacy and language difficulties [1,39]. These handbooks are retained by the caregiver so that other family members can see them.

These assessments will lead into identifying activities that can be used at home to help children move to the next stage of their development. Various manuals are available that outline a range of activities at different developmental stages, but these will have to be adapted by the support worker to suit the family [31,39,40]. Possible activities can be demonstrated to the caregiver by the support worker with the child who then watches the caregiver doing them. In addition, activities involving other adults in the family and siblings can feature. It may be that certain aids and equipment will be needed—such as standing frames or hearing aids—and the support worker can assist families in obtaining them.

Particularly advantageous for children with disabilities is having access to therapists, although they can be scarce in many locations. Yet, they can provide more detailed guidance on the learning goals suited to the child and the exercises that families can use at home. Later, we will examine strategies as to how community health services could be accessed by family-centred services.

### 7.2. Children’s Health and Nutrition

Not only do children with disabilities miss out on receiving therapy and other specialist services even when they are available in clinics and hospitals, but this exclusion extends to their physical health [41]. Children with disabilities are at greater risk of not being immunised for common illnesses and having access to medications. They are less likely to be taken to community health centres or hospitals or to see a doctor when ill. They miss out on augmented feeding programs, and many receive an inadequate diet [42]. Thus, home visitors are well placed to assess the extent to which the child’s health needs are being met as well as those of other children in the family and the health needs of the adult members. Thus, the support to families should incorporate these health needs even if there is no immediate means of addressing them. However, later sections will review possible responses. Such information can be used when it comes to lobbying for equity of access to health services [15].

More optimistically, there may be actions that can be taken by parents with support from family workers to address some of the issues. Examples include organising an immunisation session especially for children with disabilities in their locality, having a doctor hold a monthly clinic for children and families, providing a fund to cover medication expenses, and giving families advice on nutritious foods [1].

### 7.3. Identifying the Needs of the Main Caregiver

Of particular relevance to a child’s development is the caregivers’ (and others) interactions with the child and especially the need for what has been termed ‘responsive parenting’. This involves caregivers playfully interacting with the children right from birth: cuddling, smiling, and talking to them while observing and responding to children’s movements, sounds and gestures, and verbal requests. Parents may fail to appreciate the need to do this, especially with infants who have disabilities, yet there is clear evidence about the benefits it brings to all children in the preschool years and beyond, albeit the research in LMICs is limited [43]. Family support workers can model these parental behaviours in their interactions with the children as an encouragement and lesson to all family members, including mothers.

A home visit also provides an opportunity to assess the overall quality of life of the family and the personal needs of the main caregiver: how are they coping, how are they feeling? Again, there are tools available to help home visitors to open these conversations either informally by providing a listening ear, or more formally by working through a questionnaire with the caregiver [44]. Issues around stigma and discrimination because of the child’s disability will often arise. Sensitive and sympathetic listening to mothers, especially, can help them to negotiate the conflicts with cultural values and practices, and explore possible ways of dealing with them [25].

Many caregivers of children with disabilities experience mental health issues [45]. Mothers and grandmothers, in particular, are susceptible to depression and worries, lacking in energy, loneliness, and despairing about other issues, such as the abuse they may be experiencing from fathers or other family members [46]. These feelings are exacerbated when the child is especially demanding and engages in behaviours that are difficult to manage. It is more commonly recognised now that the caregivers’ emotional state has an impact on the child as well as the converse. The support worker may not be able to resolve these personal needs, but they can help families to think through how they might help themselves and by signposting other organisations to them that might assist them. In later sections, we will examine some of the other aspects of family-centred supports that may need to be developed within communities; specifically, the provision of emotional support from other caregivers can be helpful.

*Likewise, the emotional wellbeing of the child is a further consideration, although often their engagement in more activities and interactions with others is in itself a means of alleviating problem behaviours such as self-injury by biting self or hitting others. The support worker needs to empathically explore with caregivers how they are currently managing the child’s emotions and behaviours and helping them to use more positive strategies for managing the child* [47].

### 7.4. Individual Family Plans

The information gained from the above activities—which may require several visits—is recorded in a plan that summarises the main development goals identified for the child and the activities that the family will use at home. Also recorded are the actions to assist the caregiver and family. These individual family plans are a well-attested means for use in family-centred services as they provide a shared focus for family support. Simplified versions can be tailored to the family needs in LMICs [1]. More information is available at: https://ldaamerica.org/info/what-is-an-individualized-family-service-plan/ (last accessed on 1 February 2024).

The plans are also used to assess the child’s progress in terms of the goals achieved and those that require further attention. Hence the plan is regularly reviewed with families—perhaps every three months—when new goals can be identified, or existing ones modified. As we will note later, the plans are an important way of describing the range of interventions the support workers have used across the families and assessing the impact of the supports on children and families.

## 8. Mobilising Peer Support for Families

A focus solely on home-based support has also certain dangers. In particular, it devalues the contribution of support that parents can give to each other as well as failing to strengthen community connections. Furthermore, it can create an over-dependence on the family support workers which goes against the ethos of helping families to help themselves. And what will be the fall back, should the home-based support terminate through lack of funds or leadership? 

In this section, our focus is on how family-centred services can mobilise peer support as an essential complement to the home-based efforts, mainly through group meetings and the formation of informal or formalised parents and friends associations. Indeed, the latter have been internationally recognised as the major advocates for improved services locally and nationally [19]. Admittedly, there are caregivers who are reluctant to become involved in group activities for a variety of reasons, including the stigma around having a child with disabilities. However, the building of a trusted relationship with a support worker through home visits will likely increase their willingness to meet other families.

Three main benefits can derive for caregivers from being part of a parents and friends’ group that is facilitated by a family-centred support service. The term ‘friends’ indicates that other family members such as sisters and grandparents are welcome. Initially, the family support workers may organise and run the meetings but in time, these responsibilities should be shared with capable members from the group so that the group becomes theirs and its dependency on others is reduced.

### 8.1. Sharing Information and Experiences

Coming together in groups gives caregivers the opportunity to learn from one another’s experiences as well as from visiting speakers on topics of interest to them. Three types of support can emerge for group members [48]: Informational support about organisations that can assist the members, for example, and how other group members have obtained their help. Emotional support comes from hearing how others are coping with similar challenges or how they have coped in the past. Practical supports can be provided by group members such as passing on equipment and toys they no longer need or offers to look after the child in an emergency.

Training packages are available for use with groups of family caregivers which have been trialled internationally, including in low-income countries [31]. For example, the World Health Organisation has developed a Caregiver Skills Training (CST) for families of children aged 2–9 years with developmental delays or disabilities [34]. It aims to provide caregivers with skills that they can use at home to improve their child’s engagement in activities and communication, and to promote positive behaviour and skills for daily living. Caregivers attend nine group sessions, each lasting up to three hours plus three home visits at the start, middle, and end of the course. The core nine-session course has also been adapted for virtual delivery through videoconferencing for group sessions.

The teaching methods of this and other training courses encourage the exchange of emotional and practical supports among the caregivers and contribute to the benefits which caregivers derive from the course. Similar benefits can also emerge from one-off training events, as when an occupational therapist is invited to talk to the caregiver group about aids and equipment to assist children to walk or a psychologist about managing children’s behaviour.

### 8.2. Joint Advocacy

Groups can be more effective advocates for change than individuals, such as by challenging the stigma and myths around disability through meetings with local leaders and traditional healers or persuading health and education officials to provide better access to therapists and schools. Advice is available from International NGOs, Parent Associations, and Disabled People’s Organisation nationally and locally as to the strategies they have found most effective in advocating for their children’s rights. An example toolkit is available at: https://www.cbm.org/fileadmin/user_upload/Publications/Disability_Mainstreaming_Toolkit.pdf (last accessed on 1 February 2024).

A spin-off from these activities is a boost to members’ self-confidence and creating a sense of pride in having a child with disability.

### 8.3. Income Generation

Group action is also needed to alleviate the poverty experienced by many families [14,49]. In countries that provide social welfare payments, groups can ensure that their members know how to access these benefits as they play an important role in alleviating family poverty [50]. More often, though, groups may need to create their own income-generating activities for their members. These include saving schemes in which group members make regular contributions and from the savings, pay-outs can be made for members experiencing particular hardships. Also, micro-financing schemes have been successful in which loans are made to members to start their own business—such as chicken rearing—and some of the profits from the sale of eggs and meat are used to pay back the loan. There are of course risks and challenges in taking on such schemes. They are probably best considered when the groups are well established and have been successful in their other activities. Or they can join with other community groups to participate in their schemes as the following section describes.

## 9. Mobilising Community Support

A further dimension of family-centred supports is to assist caregivers to access whatever services are available in their locality and also making them more suited to meeting the needs of the child with disabilities. This can be achieved through promoting family and group advocacy as noted above, but the leaders may also need to be proactive in facilitating change within local communities [9,33].

Among those groups that deserve particular attention are the following.

### 9.1. Traditional Healers and Faith Communities

Families often turn to traditional healers in seeking a cure for their child’s disabilities. However, the healers often perpetuate mistaken beliefs that increase the self-stigma families experience [51]. Nevertheless, traditional healers and faith communities may have a role to play in providing emotional support to caregivers as mothers often acknowledge that their religious or spiritual beliefs are an important coping strategy for them [52]. Examples are available of dialogues that have been undertaken with traditional healers and faith communities and the more positive contribution they could make to families [7,53]. Overall, though, the engagement of traditional healers and leaders remains an unresearched area.

### 9.2. Health and Social Services

Mention has been made above of how health and social services need to be made more accessible for children with disabilities [41]. Family-centred support services have advocated for clinics to hold an immunisation session especially for children with disabilities in their locality and organised health workers to provide training sessions for families on nutritious foods. In return, the family support services have provided places for community health workers on their training courses to make them more aware of the early signs that a child may have some developmental difficulties and activities they can recommend to parents to promote the children’s development.

Ideas can be shared with managers and healthcare staff on the adaptations that could be made to their clinics to ensure the health needs of preschoolers with disabilities are met. Others have negotiated for hospital-based therapists to undertake an outreach clinic in a local community facility. Likewise, arrangements have been made for hospital paediatricians or those in training, to visit the locality once a month [1]. Issues around the cost of medications for children with epilepsy or appliances for children may need to be resolved, perhaps through charitable fundraising.

### 9.3. Poverty Alleviation

Families of preschoolers with disabilities also need to be included in any poverty alleviation schemes that are active in the locality such as employment training and grant and saving schemes [49]. As noted above, family support workers can make introductions to the scheme and offer advice and guidance on how any additional needs of the families could be accommodated within the service. More problematic is instigating income-generating schemes in localities where none exist. It could be that parents and friends’ groups could take on this task, albeit on a small-scale basis as described above. If running successfully, this will strengthen the advocacy for a more expanded service that can benefit many more families with and without a child with disabilities.

### 9.4. Approaching Community Groups

There may be other groups which may need to become better informed about the needs of families who have a member with disabilities, such a minibus drivers and local politicians. But with these community groups and those listed above, there are common approaches that have been effective.
Engage the target community in conversations based around stories of the children and their progress: how they and their families have been helped. Give them the opportunity to meet some of the children and their caregivers. Pose the question: what could you do to help?Offer your assistance to them and assure them that you envisage a partnership that will be mutually beneficial.Identify the potential change leaders in the community and build a trusted relationship with them.Positively affirm their efforts to include the children and give them the credit for their achievements.

## 10. Preschool Education

The value of providing group-based preschool, nursery or kindergarten education for children is well recognised internationally and has become a major feature of government policy in high-income countries. The benefits include children’s social interactions with their peers, access to a structured and stimulating curriculum, familiarisation with school routines, and preparation for entry to primary schools. Equally, caregivers are enabled to join the workforce and generate income for the family or engage in other pursuits, such as vocational training.

Preschoolers with disabilities and their families can equally benefit from enrolling in centre-based education. However, in many low-income countries preschool education is not widely available for children without disabilities and even less so for those with a disability. Moreover, the push to provide education for children with disabilities in low-income countries has been focussed on primary schools with little attention paid to preschool provision [54]. In recent years, strategies have emerged for the successful inclusion of children with disabilities in local schools [55,56]. Arguably, these are applicable to any preschool centres that are currently available in a locality, or they can guide the setting up of new facilities. For all these reasons, family-centred services need to incorporate options for centre-based educational provision for the children while still maintaining the ethos of family-centred support. Two main options are available: placements in existing centres and creating new facilities.

### 10.1. Partnering with Existing Centres

Existing centres can be visited to determine the activities they provide, the accessibility of the premises, the staff employed, and crucially their attitudes or experience of enrolling children with disabilities. The main reservations tend to be around lack of knowledge and the skills of staff in managing such children and the concern that other parents might object to their child mixing with a boy or girl with disabilities. Yet, these can be balanced by a respect for a child’s right to education and the benefits it brings along with a willingness to trial a placement, especially if a sibling has already or is attending the centre. An offer from the family support service to assist the centre may be welcomed. For example, the family support workers can become a resource for centre staff as they share their knowledge of the child and the ongoing learning activities and adaptations being used at home by caregivers. In addition, staff can be invited to attend training courses provided by the family support service. Suggestions for handling the reactions from other parents can be shared, primarily by highlighting supportive reactions from some if not most of the parents. But some issues might be more intractable: the family’s ability to pay the fees, for example.

Arguably, these strategies apply equally when children come to primary school age and their enrolment should be considered as an integral part of the overall planning of the service as a further contribution to community development [1].

### 10.2. Creating Preschool Educational Opportunities 

Options for developing preschool education for children and caregivers could be explored. For example, some of the time of the family support workers could be redirected by starting a preschool experience for a small group of children with disabilities and any siblings of preschool age. Caregivers would also be expected to take part on a rota basis and the support workers might be assisted by other volunteer helpers from the locality. The group might meet in a community facility such as a faith centre for two to three hours on selected days of the week.

More ambitiously, the centre might seek funds to employ dedicated staff—ideally, nursery trained teachers—to provide a preschool education for children with disabilities, although this could also serve preschoolers without disabilities so that an inclusive environment is provided. Again, partnerships with other community groups or organisations might assist with creating new services in the locality. Examples of preschool curricula are available to guide the staff along with online training courses. Examples are given at: Best Early Childhood Education Courses Online with Certificates [2024] | Coursera (Last accessed on 1 February 2024). Initiatives such as these, when established with evidenced benefits, can then be used to advocate for governments to fund the costs of the centre, notably the salaries of staff.

If preschool education options are not feasible, energy will still need to be directed towards the children attending primary schools when some of the above strategies alongside others may need to be employed. Here, too, school–family partnerships will be crucial to its success [57].

## 11. Documenting and Sharing the Impact of the Service

The preceding sections have illustrated the diversity of tasks facing the leaders of family-centred services for preschoolers with disabilities. But there is yet another important function of the leaders, namely, documenting the activities of the support services provided, the lessons learnt, and the impact they have had on the children’s development, their inclusion in their family and community, as well as the outcomes achieved for caregivers and their families. All of this serves the very important purpose of adding to an international understanding as to how underserved and disadvantaged communities can be supported. In that respect, this paper is an example of how the current literature can guide contemporary efforts and perhaps inspire others to embark on responding to the needs in their country [58].

Nonetheless, leaders may be so busy delivering the supports to families that they do not have the time and resources to write about their work. Devising systems for keeping essential records and compiling reports may ease the burden, especially by using information processing software available on tablets and smart phones. These include: Creating a database that collates the pertinent details. of the children and families who avail of the service.Documenting the various supports availed of by families: the number of home visits, attendance at group meetings, training events provided, and so forth.Providing a computer-based pro forma for documenting individual family plans that can be updated in terms of goals achieved for both the child and the caregiver.Maintaining a diary of meetings held and the outcomes from them, especially with respect to mobilising communities.Obtaining feedback from families about the supports they have been given. This can be achieved through interviews with a sample of families and/or in group meetings.Organising celebrations of significant achievements—such as parties and concerts—for families, support staff, and volunteers.Keeping financial accounts of income and expenditure. This will assist in making estimates of the cost benefits arising from the supports provided.The above information can be brought together into an annual report accompanied by photographs and infographics highlighting the main outcomes from the support service. The report can be shared with print and broadcast media to inform local communities and provide positive news stories to counter the stigma around disability. The report can be used to lobby decision-makers in local and national government for resources to further develop community supports.

Leaders of local services can also act as advisers to personnel from other areas or countries who want to start similar services. This could be achieved through visits, online meetings, and the sharing of resources developed by the service, such as training materials for use with caregivers or in preparing family support workers.

### Planning for Sustainability

The above activities will contribute to the sustainability of the service, but the sad fact is that these forms of community initiatives often collapse despite the beneficial impacts they have brought to children, families, and communities. The chief reason is a lack of resources—primarily, the cessation of funding from donors or governments alongside the departure of leaders and other significant personnel.

Creating proactive plans to manage these risks is essential, such as reducing the dependency on one funding stream coupled with nurturing future leaders and support workers. And there are other factors that threaten the continuity of services—such as natural disasters and civil unrest—which cannot always be anticipated and that may well result in the termination of the service. That too is a lesson for others and in that respect, it should not be seen as a failure but rather the leaders should take comfort for the good that was done and the learning it generated.

## 12. Conclusions

The needs of preschoolers with disabilities and their families are well documented internationally. Rights statements in country constitutions and by the United Nations along with policy documents and national implementation plans have had little impact on meeting their needs in low-income countries over the past 30 years. Yet, we know what could be done to provide them with a better future. Family-centred, locality-based actions have been shown to be effective even in the most impoverished communities. And they are not confined to low-income nations. Arguably, there are lessons here for service delivery in more affluent countries, especially for immigrant families who have preschoolers with disabilities [59,60].

The knowledge and the strategies described in this article provide a road-map for others to follow. Yet, translating written words into practical actions is but a first step. And it can begin with just a few people; as Margaret Mead opined: “Never doubt that a small group of thoughtful, committed citizens can change the world; indeed, it’s the only thing that ever has”.

## Figures and Tables

**Figure 1 ijerph-21-00651-f001:**
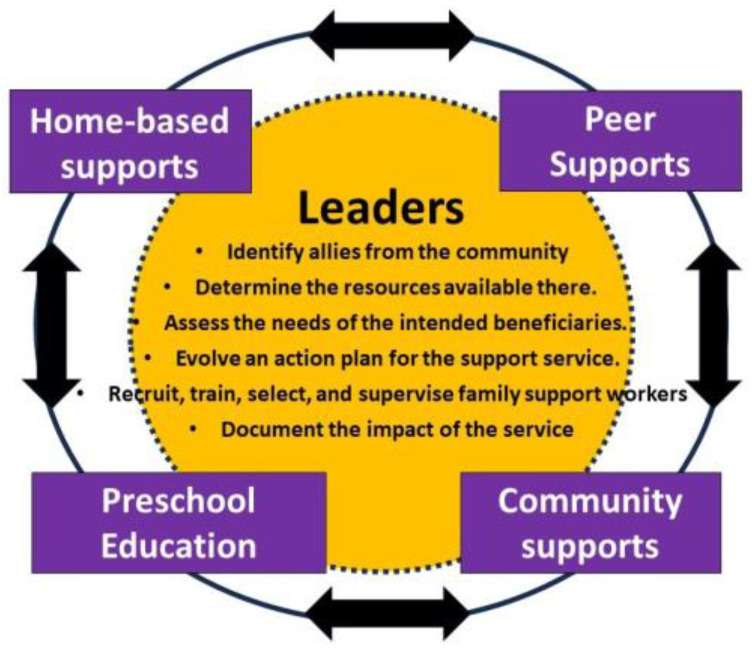
Key elements of family-centred supports.

## Data Availability

Not Applicable.

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
