# Peer review of "Creating Family-Centred Support for Preschoolers with Developmental Disabilities in Low-Income Countries: A Rapid Review to Guide Practitioners"

_ijerph, 2024, doi:10.3390/ijerph21060651_

Round 1
Reviewer 1 Report
Comments and Suggestions for Authors
Dear Author
Many thanks for submitting this manuscript for peer review. Although of potential interest to the readership of this review. I have one major flaw that I feel is of appropriate grounds to reject the paper in its entirety. You mention that this is a rapid review of the literature. However, I see no evidence of a review taking place, bar briefly in Appendix A. There needs to be an appropriate, academically written introduction, providing context as to why such a review was necessary, then a detailed methods showing what aspects of a structured review was taken and applied to this review, with rationales given to support your claims. Additionally, from the piece of the review that I read, you seem to use the first person quite often, which unless a clear rationale as to why you did so is present, this should not occur. As such, my advice is either add these elements into the paper to make it a rapid review or amend accordingly to take reference to a rapid review out and leave this as a perspective piece that can then be resubmitted for review as a perspective piece by the journal.
I hope this is helpful.
Author Response
Note: The reviewer's comments are in italics and the author's response in plain text; including quotations from the submitted paper.
However, I see no evidence of a review taking place, bar briefly in Appendix A. There needs to be an appropriate, academically written introduction, providing context as to why such a review was necessary, then a detailed methods showing what aspects of a structured review was taken and applied to this review, with rationales given to support your claims.
Many thanks for reading this paper but perhaps I did not make sufficiently clear that this was not a paper aimed at academics but rather an attempt to highlight for practitioners the main lessons to emerge from past research studies. In short the focus was on the implementation and dissemination of public health research. Hence I had written at the outset: "In this review, my aim is to capture the practical steps that can be taken by practitioners in any or every country to enhance the life chances of youngsters with developmental disabilities, primarily through family-centred, family-driven supports. Nor is this intended to be an academic endeavour. Rather my message is for practitioners – be they family members, community workers, health professionals – who are often eager to assist but feel they lack the knowledge and skills to do so"
I wonder too if you paid sufficient attention to the text in the Appendix (I chose to do this as I had not intended the details of the review to be foregrounded in the paper). In the Appendix, I provided the information you stated needed to be included. Here I gave the criteria for selecting papers: "priority was given to identifying papers published in the last five years that could be accessed by practitioners free of charge along with websites from which resources could be downloaded"
I noted the main search engines used: "Google Scholar was initially searched and cross-checked with other search engines such as Web of Science. In addition, the reviews were augmented with pertinent individual articles and reports from authoritative organisations to elaborate on the issues identified in the reviews. Some of these were initially identified using Microsoft Bing AI".
I also described the search terms used: “preschool”, “disability”, “family-centred”, “Low Income”; “Review”
I provided a table listing the review articles identified: "Table A1 below gives details about the selected review papers, all of which are listed in the references with a doi number provided to facilitate reader’s internet access to each review paper. In total, over 1,500 studies were included in the 23 identified reviews".
Additionally, from the piece of the review that I read, you seem to use the first person quite often, which unless a clear rationale as to why you did so is present, this should not occur.
I used the word 'I' twice. Did you not read the rationale that I gave for so doing?: "I have been involved in the research and development of family-centred supports in the UK and Ireland for over 50 years and since the mid 1980s; starting with Zimbabwe and since then, extending to other African countries, to the Asia-Pacific Region and Guyana in South America".
As such, my advice is either add these elements into the paper to make it a rapid review or amend accordingly to take reference to a rapid review out and leave this as a perspective piece that can then be resubmitted for review as a perspective piece by the journal.
I note your advice but I leave it to the editor to judge whether your review is a fair assessment of the paper. Your judgement seems to be based on a hasty reading and a preconceived perception that the Journal is solely to be read by academics and researchers.
Reviewer 2 Report
Comments and Suggestions for Authors
Thank you for the opportunity to review “Creating family-centred support for preschoolers with developmental disabilities in low-income countries: A review to guide practitioners (ijerph-2967890). I appreciate the author’s efforts to offer a new style of review that may serve as an example for public health researchers and academics for a “more rapid implementation of evidence-based knowledge into existing and new support services.” While innovative in its approach, the author presents an interesting and valuable avenue of which to disseminate existing research to practitioners, and direction to researchers and educators in the field to use the content in training endeavors.
Overall, the manuscript focuses on an important issue- supports for preschool children with disabilities in low-income countries. It provides key knowledge and strategies that I believe make a positive contribution to the field. Below, I describe a couple areas of concern, and opportunities to strengthen the manuscript as it is currently written.
· Section 5 describes what is meant by family-centered supports. A list of key principles and values that are expressed in building respectful and trusted relationships between support workers and parents is included. I was surprised that something about cultural responsiveness is not on this list. That is, that a respectful and trusted relationships between support workers and parents includes consideration of family’s culture, beliefs, and values.
· Appendix A provides a brief explanation as to how the review was conducted. The author reviewed 23 identified reviews (with over 1,500 studies included). Although the author provides a table of the 23 selected reviews included in the paper, the author states “I judged the relevance of the articles to be included in this review which is why it is ‘termed’ rapid and possibly incomplete” (line 736). Please provide further details as to how the articles were judged as relevant.
Author Response
I appreciate the author’s efforts to offer a new style of review that may serve as an example for public health researchers and academics for a “more rapid implementation of evidence-based knowledge into existing and new support services.” While innovative in its approach, the author presents an interesting and valuable avenue of which to disseminate existing research to practitioners, and direction to researchers and educators in the field to use the content in training endeavors.
Many thanks for your endorsement and the value of this type of review especially in open access journals which have the potential to reach practitioners globally at no cost.
Overall, the manuscript focuses on an important issue- supports for preschool children with disabilities in low-income countries. It provides key knowledge and strategies that I believe make a positive contribution to the field. Below, I describe a couple areas of concern, and opportunities to strengthen the manuscript as it is currently written.
Again your endorsement is much appreciated of the importance of addressing the omission in public health policies of preschool children with disabilities and their families in low-income countries. My thanks too for your helpful suggestions.
Section 5 describes what is meant by family-centered supports. A list of key principles and values that are expressed in building respectful and trusted relationships between support workers and parents is included. I was surprised that something about cultural responsiveness is not on this list. That is, that a respectful and trusted relationships between support workers and parents includes consideration of family’s culture, beliefs, and values.
You are right and I am embarrassed by this omission. I have added a further point in section 5 with a new reference added.
“ ....to which can be added respect for family cultures, values and beliefs, which is especially pertinent in non-Western countries and for immigrant families from these cultures [25].”
- Jansen-van Vuuren, J., Aldersey, H.M. Stigma, Acceptance and Belonging for People with IDD Across Cultures. Curr Dev Disord Rep 7, 163–172 (2020). https://doi.org/10.1007/s40474-020-00206-w
Also in section 7.3 Identifying the needs of the main caregiver., these words have been added:
“Issues around stigma and discrimination because of the child’s disability will often arise. Sensitive and sympathetic listening to mothers especially, can help them to negotiate the conflicts with cultural values and practices, and explore possible ways of dealing with them.[25]”.
Appendix A provides a brief explanation as to how the review was conducted. The author reviewed 23 identified reviews (with over 1,500 studies included). Although the author provides a table of the 23 selected reviews included in the paper, the author states “I judged the relevance of the articles to be included in this review which is why it is ‘termed’ rapid and possibly incomplete” (line 736). Please provide further details as to how the articles were judged as relevant.
I have reworded the paragraph as follows.
"The relevance of articles to be included in this review of reviews was based mainly on the search terms noted above, but also on publication in the last five years, availability for free download and containing implications for practice. It was a rapid review by one person and hence possibly incomplete."